# Scoping Review of Acupuncture and Moxibustion in the Treatment of Inflammatory Bowel Disease

**Affifa Farrukh *** and **John Francis Mayberry**

Department of Gastroenterology, Nuffield Hospital, Scraptoft Lane, Leicester LE5 5HY, UK
* Correspondence: farrukh_affi@yahoo.com

**Abstract:** Acupuncture and moxibustion are complementary therapies used by a significant number of patients with inflammatory bowel disease. There is limited research evidence of their effectiveness in the management of ulcerative colitis and, albeit less limited, in the case of Crohn's disease. However, due to a lack of knowledge, their use as additional supportive therapy by gastroenterologists and specialist nurses is uncommon. Current evidence would indicate that they have a place as additional supportive treatments for patients with inflammatory bowel disease and their efficacy should be assessed through appropriately powered trials. In days of shared care and responsibilities between patients and clinicians, there is a need to share such information with patients so that they can make informed decisions.

**Keywords:** Crohn's; ulcerative colitis; acupuncture; evidence

## 1. Introduction

Both forms of inflammatory bowel disease, namely Crohn's disease and ulcerative colitis, have become more widespread and common during the last 50 years. The aetiology of both conditions remains unclear, with a range of causes having been postulated over that time. As a consequence, treatment has been directed at the control of symptoms rather than curing the disease. Even with the advances associated with the introduction of biologic therapies, the conditions remain chronic and characterised by recurrent flare-ups. The impact of these diseases on day-to-day life is significant and the unexplained symptom of tiredness probably plays a significant role in these effects. As a result, the use of complementary therapies by patients with inflammatory bowel disease is common and well recognised. For example, in a population-based study from Manitoba, Canada, it was suggested that 3.5% of patients have acupuncture treatment [1]. However, the figures for acupuncture therapy have been as high as 33% from France [2] and Germany [3]. In Hungarian patients, it is used by younger patients, those with higher educational levels, and those who are taking immunosuppressant therapy. In this community, it tended to be associated with poorer compliance with allopathic treatments [4] and probably reflects a dissatisfaction with their efficacy.

It is against this background that this review of published work on the potential value of acupuncture and moxibustion in the clinical management of ulcerative colitis and Crohn's disease is undertaken. This review does not consider the mechanism by which they may be effective, particularly as many such studies have been conducted on rodents. Rat and mice models of inflammatory bowel disease are chemically induced. Therefore, their applicability to the human condition is questionable in that Crohn's disease has not been described in animals and naturally occurring colitis has been largely confined to dogs [5]. A significant number of studies on the role and effectiveness of acupuncture in such animal models have been published during the last decade. In contrast, the number of studies on patients has been limited during this period.

Various techniques have been used in the treatment of inflammatory bowel disease. Within the acupuncture option, traditional Chinese therapy, electroacupuncture and catgut

embedding at acupuncture points have all been investigated. Traditional Chinese acupuncture uses non-cutting thin needles and normally requires two or three sessions of treatment before any improvement in clinical symptoms can be seen. Its philosophical basis of balancing energy flow around the body has been the subject of significant criticism by many exponents of evidence-based practice. However, there is an extensive body of literature demonstrating its efficacy in a wide range of chronic diseases. Although treatments are tailored to individual patients, most practitioners will use similar points. During a treatment session, acupuncture needles are inserted into the skin at right angles to the surface and to various depths. They are generally left in place for between 30 and 45 min. Electroacupuncture uses the same needles and points of insertion. However, this is a modern variation which aims to enhance the benefits of traditional treatment. Two needles are placed at each acupuncture point and small amounts of electricity are passed through the needles to achieve a vibration effect, comparable to that obtained by physically moving them, as happens with the twirling of the needles in traditional therapy. The principle behind catgut embedding at the acupuncture points is to achieve prolonged stimulation and to reduce the frequency of treatments and is a technique that was developed in the second half of the twentieth century.

Moxibustion is a form of treatment, which like traditional acupuncture, has existed for thousands of years. It may be offered in combination with acupuncture or as a separate treatment and appears to be commonly used in many NHS obstetric units to complement conventional treatments for the management of breech presentations. With direct moxibustion, cones of moxa, prepared from the ground young leaves of Artemisia vulgaris, are placed on acupuncture points and lit so as to smoulder. Care needs to be taken that such heated cones do not cause blistering of the skin in Western practice. In order to avoid such issues, indirect moxibustion is more commonly practiced in the West. With indirect moxibustion, acupuncture needles are warmed with moxa and with partitioned moxa; a herb, such as ginger, is placed between the moxa and the patient's skin. An alternative approach in Western practice is to use commercial sticks of smouldering moxa which are brought close to the skin at acupuncture points with a pecking motion, the frequency of which is determined by the patient's experience of uncomfortable heat.

Personal series and case reports have been largely excluded from this review. Its purpose, rather, is to identify potential forms and points of treatment which could be applied as adjuvant therapies alongside conventional treatment. Studies on the use of complementary therapies by patients with inflammatory bowel disease have shown that there is a readiness to both consider and use them. By bringing together the evidence-based treatment programs for these conditions, it is to be hoped that they will become more widely available and could be easily used to ameliorate some of the effects of these chronic conditions.

## 2. Ulcerative Colitis

Ulcerative colitis is limited to the colon and found most commonly in non-smokers or ex-smokers. It is a chronic condition for which there is no cure and there is an increased risk of colorectal cancer [6]. For most of the last century, the cornerstone of treatment has been 5ASA compounds [7] and there is some evidence that they can significantly reduce the risk of colonic carcinoma [8]. Where 5ASA compounds have failed, immune suppressants, such as azathioprine, and biologics are commonly used. However, for the management of flares, steroids are still the drug of choice. Despite this range of therapies, when they are ineffective, surgery, in the form of a colectomy, remains the only generally accepted treatment option. It is against this background that a significant proportion of patients seek other therapies, including probiotics, fish oils and herbal therapies [9]. Cultural background appears to have little influence in this decision, with European and South Asian patients equally seeking help from such remedies [10].

### 3. Acupuncture and Moxibustion in Ulcerative Colitis

In a meta-analysis of 13 randomised controlled trials, Wang et al. [11] showed that both acupuncture alone and acupuncture combined with conventional medicine were more effective in treating ulcerative colitis when compared to conventional medicine. This is true for both manual acupuncture and electroacupuncture. Similarly, moxibustion has been shown to be probably effective in a metanalysis [12]. However, many studies of both acupuncture and moxibustion are criticised on the grounds that the diagnostic standards and efficacy criteria, which had been used, were neither uniform nor standardised [13].

### 4. Techniques and Choice of Points

In a study of 62 patients with ulcerative colitis, acupuncture and moxibustion were applied at ST 25 and REN 4 in one group, whilst the other group received conventional medical treatment. Acupuncture and moxibustion were shown to be effective treatments [14].

The efficacy of acupuncture combined with acupoint catgut embedding sequential therapy in mild to moderate ulcerative colitis was investigated in a study of 120 patients, who were randomized into an acupuncture and acupoint catgut embedding sequential therapy group or a mesalazine group. In the acupuncture group, needling was performed at ST 25, ST 37 and LI 11 during the active phase and acupoint catgut embedding was applied at BL 20, ST 36 and CV 4 during the remission phase. Treatment with either modality was given for 12 weeks. After treatment, symptom scores improved in both groups, but the effect from acupuncture was significantly better than that from mesalazine. In addition, recurrence following acupuncture was 8.5% (4/47), which was significantly better than the 32.4% (11/34) figure seen amongst those treated in the mesalazine group ($z = -2.7$, $p < 0.006$) [15] (Figures 1 and 2).

Traditional catgut is made from collagen taken from the serosal or submucosal layer of the small intestine of cattle, sheep and goats. Acupoint catgut embedding therapy has been investigated in a study of 116 patients of whom 56 cases had catgut embedding at BL 25, ST 36, ST 37 and other points. The control group received sulfasalazine at a dose of 4 to 6 g/day. Symptomatic and endoscopic outcomes were better in the catgut group at 8 weeks [16]. However, its use in many countries has been banned because of the risk of animal to human disease transmission. Traditional materials have been associated with localised reactions, including granulomas [17]. It is unknown whether the synthetic absorbable polymers, which have replaced it in surgical practice, would have the same clinical effects. Indeed, catgut acupuncture has a short history, first being introduced as a novel therapy in the 1970s [17]. Its concept is aligned with that of Press Tack Needle (PTN) therapy, whose objective is to prolong the duration of response to acupuncture over several days [18]. Although there have been no studies of PTN in the management of inflammatory bowel disease, it has the potential to be used as a home therapy in the early stages of a flare-up, but this would require rigorous assessment in therapeutic trials alongside conventional treatment.

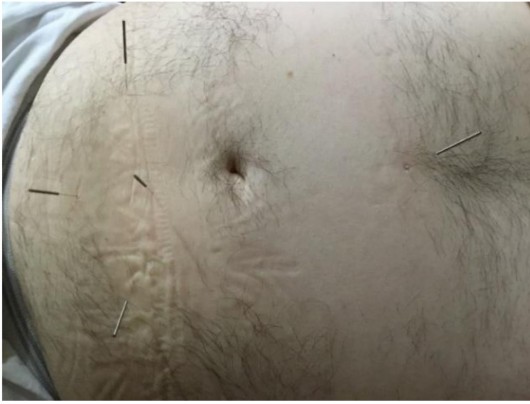

**Figure 1.** *Cont.*

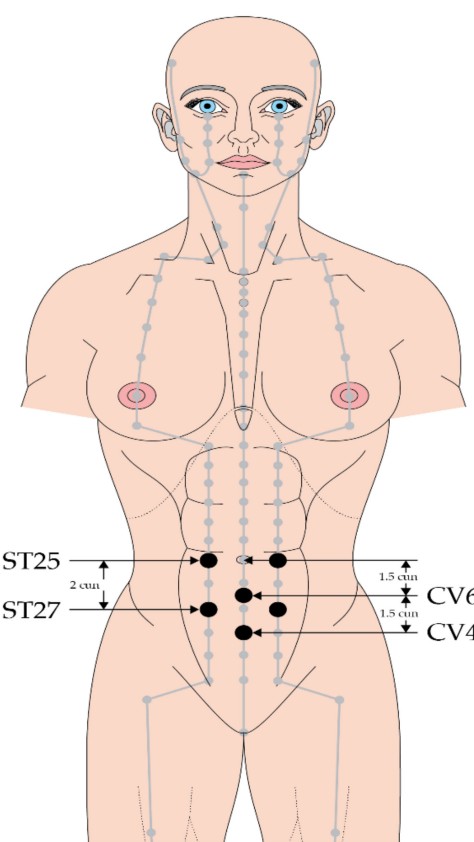

**Figure 1.** Abdominal Acupuncture at CV4, CV6, CV12 & ST27 (Measurements are in Cun—the width of a thumb).

Three treatments with warm needle moxibustion for 30 min at ST25, CV4 and CV12 have been shown to be superior to acupuncture and to treatment with sulfasalazine and azathioprine in a study of 117 patients. Patients who received warm needle moxibustion had a 15–18% better outcome than the other two groups [19] In warm needle moxibustion, cones of moxa are attached to the end of the acupuncture needle and lit. The smouldering moxa heats the needle and is, therefore, a form of indirect moxibustion (Figure 3).

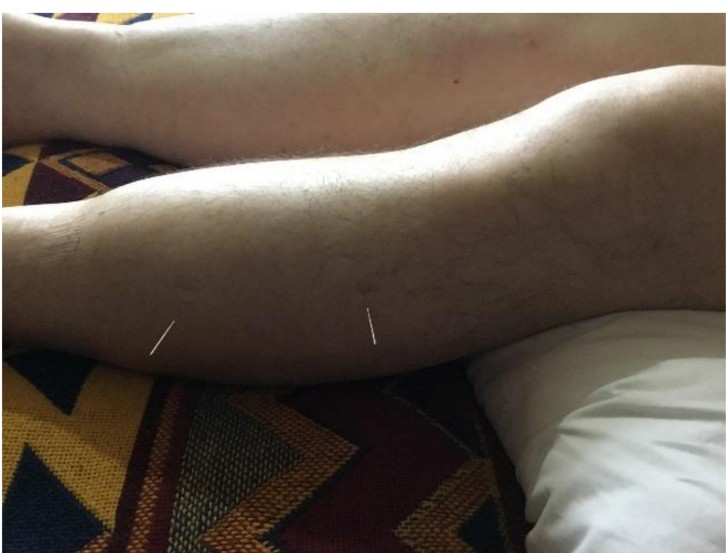

**Figure 2.** *Cont.*

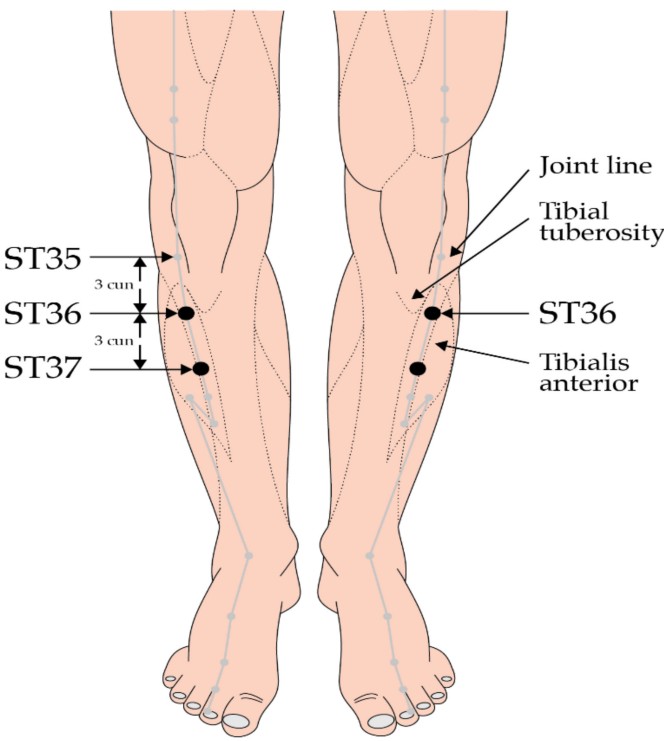

**Figure 2.** Acupuncture at ST36 and ST37 (Measurements are in Cun—the width of a person's thumb).

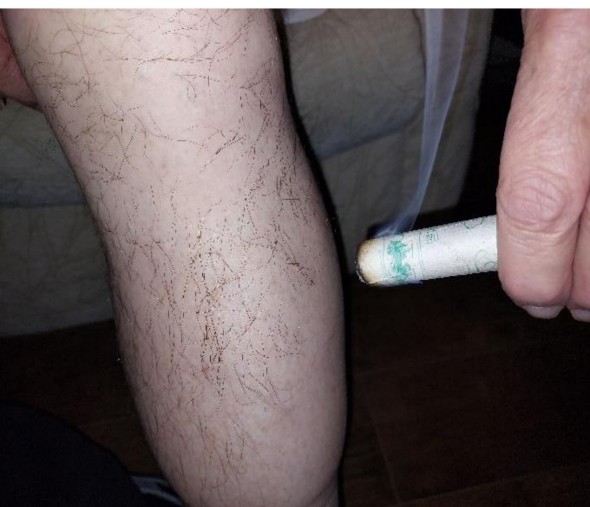

**Figure 3.** Moxibustion at ST36.

Use of moxibustion is associated with the production of smoke containing chemical agents [20] and this raises the possibility that its effect on colitis may be comparable to that of smoking tobacco [21]. For example, moxa smoke has been shown to contain a variety of complex components, including ammonia, alcohols (ethylene glycol, pentyl butanol), aliphatic hydrocarbons, aromatic hydrocarbons, terpene compounds and their oxides amongst others [22].

A recent analysis of the potential risks from burning moxa sticks found 112 toxic compounds present and the researchers advocated that moxibustion rooms should have enough artificial ventilation to ensure the health of both patients and practitioners [23]. It is of some interest that a protective and treatment role for potentially toxic compounds has been shown for betel nut [24] and arsenic [25], respectively, in addition to tobacco [26].

## 5. Difficulties with Study Design

Studies of the effectiveness of both acupuncture and moxibustion are plagued by the lack of adequate controls. Sham acupuncture has been repeatedly shown to have greater impact than conventional placebos in randomised controls of drug therapy. For example, in a prospective, randomized, controlled clinical trial of 29 patients with mild-to-moderately active UC, in which traditional acupuncture and moxa were compared to sham acupuncture consisting of superficial needling at non-acupuncture points, the Colitis Activity Index decreased from 8.0 to 4.2 points compared to 6.5 to 4.8 points in the control group [27]. Although this difference reached significance, both groups experienced significant improvements in general well-being and quality of life. The conclusion was that both traditional and sham acupuncture seem to offer an additional therapeutic benefit in patients with mild-to-moderately active UC [15]. This finding underlines the issues of sham needling [28].

One approach to such difficulties is to compare acupuncture with conventional treatment. Jia et al. [29] undertook such a study. Electroacupuncture was given at CV12, ST25, CV4 and alternated with acupoint sticking therapy at ST37, BL20, BL25 and ST36. The control group was treated with oral mesalazine. Patients who received acupuncture treatment were significantly better than those treated with mesalazine and this included endoscopic assessments [29].

Where concerns about the ethics of such a comparison arise the study design could be of acupuncture + conventional treatment and conventional treatment alone. Such a design would identify any additional benefit from acupuncture or moxibustion. It would also allow acupuncture treatment protocols to be tailored to the needs of individual patients.

The interpretation of many studies on the role of acupuncture and moxibustion in the treatment of ulcerative colitis is further limited by failures to

1.  Monitor the efficacy of treatments for sufficient periods. Ulcerative colitis is a chronic condition and comparison of responses at 4 or 6 weeks is far too short a period to allow meaningful clinical assessments to be made. Related to this need for more prolonged trials is that acupuncture, like allopathic medications, is not a cure but needs to be administered on a regular basis. The frequency of such treatments awaits the outcomes of further research.
2.  Identify clear measurable endpoints, including the use of internationally recognised clinical assessment tools and patient related outcomes. Studies require pre and post-treatment endoscopy assessments with biopsies.
3.  Include and document patients with comparable severity of disease in intervention and control groups. Trials should include sufficient cases so that they have adequate power to give clinically meaningful results.

One of the few studies which attempted to meet most of these criteria was a comparison of herb-partitioned moxibustion at CV6, ST25 and ST37 with sham moxibustion. Patients who received active treatment showed a significant improvement in recognised clinical and biochemical scores. However, both active and sham groups showed improvement in the histology of biopsy samples [30].

## 6. Crohn's

Crohn's disease is a condition which emerged in the first half of the 20th century and has subsequently spread worldwide. Its aetiology is unknown. Treatment remains symptomatic with the same agents used in ulcerative colitis, although with different efficacies. As with ulcerative colitis, there is a significant risk of colorectal cancer [31].

## 7. Crohn's Disease, Acupuncture and Moxibustion

In a study of 51 patients from Erlangen, Germany, patients who received acupuncture showed an improvement in their Crohn's Disease Activity Index (CDAI) of 97 points with

10 sessions over a period of 4 weeks and followed up for 12 weeks; the control group receiving sham acupuncture increased only 39 points [32].

In 2014, Bao et al. reported a randomised control trial in 92 patients who received either herb-partitioned moxibustion combined with acupuncture, or as a control group wheat bran-partitioned moxibustion with superficial acupuncture at a point some distance from those used in the active group [33,34]. Partitioning with herbs or wheat ensured that moxa was not directly applied to the skin and, therefore, was again a form of indirect moxibustion. The points of acupuncture were ST36, ST37, SP4, SP6, KI13 and LR3. Moxibustion was applied at ST25, CV6 and CV12. The patients had three treatment sessions per week for 12 weeks and were followed up for 24 weeks. The Crohn's Disease Activity Index (CDAI) scores improved in both the active and control groups with treatment, but more so in the active group where the improvement was also maintained during follow-up. Thirty-two of forty-three patients in the active treatment were considered to be in clinical remission at the end of the study compared to only fifteen of forty-two in the control group. Although there was no evidence of endoscopic improvement, tissue histology had significantly improved in the active group. In a separate report of 66 patients who were followed for 48 weeks, there was a statistically significant improvement in the CDAI, in the Crohn's disease endoscopic index of severity (CDEIS) and the histopathologic score in the patients who received active treatment.

The acupuncture points used in these studies are summarised in Table 1 and the studies themselves in Table 2.

**Table 1.** Points used in Acupuncture and Moxibustion Treatments.

| | Acupuncture | Moxibustion | Catgut Embedding |
|---|---|---|---|
| **Ulcerative Colitis** | | | |
| | ST25 | ST25 | |
| | ST37 | ST37 | |
| | REN4 | | |
| | LI 11 | | |
| | | CV4 | |
| | | CV6 | |
| | CV12 | CV12 | |
| | BL20 | | BL20 |
| | BL25 | | BL25 |
| | ST36 | | ST36 |
| | | | ST37 |
| | | | CV4 |
| **Crohn's Disease** | | | |
| | ST36 | | |
| | ST37 | | |
| | SP4 | | |
| | SP6 | | |
| | KI 13 | | |
| | LR3 | | |
| | | ST25 | |
| | | CV6 | |
| | | CV12 | |

**Table 2.** Summary of Studies.

| Authors | Therapy | Type of Study | Points |
|---|---|---|---|
| **Ulcerative Colitis** | | | |
| Wang et al. [11] | Acupuncture | | Metanalysis |
| Lee et al. [12] | Moxibustion | | Metanalysis |
| Yang & Han [14] | Acupuncture + Moxibustion | Trial | ST25 REN4 |
| Wen et al. [15] | Acupuncture + Catgut embedding | Trial | ST25 ST37 L11 BL20 ST 36 CV4 |
| Li et al. [16] | Catgut embedding | Trial | BL25 ST36 ST37 |
| Huang [19] | Warm needle moxibustion | Trial | ST25 CV4 CV12 |
| Jia et al. [29] | Conventional and electroacupuncture | Trial | CV12 ST25 CV4 ST37 BL20 BL25 ST36 |
| Qi et al. [30] | Moxibustion | Trial | CV6 ST25 ST37 |
| **Crohn's disease** | | | |
| Bao et al. [33,34] | Moxibustion + acupuncture | Trial | ST36 ST37 SP4 SP6 K113 LR3 ST25 CV6 CV12 |

## 8. Various Styles of Acupuncture

Most research on the role of acupuncture in the management of inflammatory bowel disease has utilised traditional Chinese points. There are no critical studies of other styles and the only support for their use comes from anecdotal accounts of their efficacy. Such reports are frequently derided by those seeking an evidence base for any clinical practice. Any trial which will meet the criteria of adequate power, appropriate randomisation and effective blinding of assessors requires adequate funding to cover the expenses of clinicians' time, detailed documentation and analysis together with insurance cover for the participants. This is a major issue and has limited studies assessing the role of traditional acupuncture.

It is even more apparent for other schools, such as Master Tung. This school of acupuncture has become increasingly popular in the West in recent years. It makes maximum use of points on the upper and lower extremities and the head, and does not traditionally use moxibustion [35].

As yet, there have been no comparative studies of Master Tung acupuncture with allopathic treatment of inflammatory bowel disease. This is also true for other schools, including auriculoacupuncture, Dr Tan's Balancing Acupuncture and Korean-stye acupuncture. This contrasts with traditional acupuncture for which there is a modest supporting evidence base for its use alongside allopathic treatments for inflammatory bowel disease.

## 9. Conclusions

Patients with inflammatory bowel disease have chronic conditions which require life-long medical treatment and often demand significant surgical interventions. It is, therefore, not surprising that a significant proportion of patients seek alternative forms of therapy so as to improve their quality of life. This often includes attention to diet and use of herbal remedies. Acupuncture offers a different approach, often specifically tailored to individual patients. There is clear, if limited, evidence confirming the efficacy of both acupuncture and moxibustion in the management of inflammatory bowel disease (Table 2). In an era when clinicians should provide patients with comprehensive information about effective treatments and, within the UK, the impact of *Montgomery v Lanarkshire* has been to make this mandatory, it is important that patients with inflammatory bowel disease are told of the potential benefits of acupuncture [36]. In order to make this a reality, gastroenterologists and inflammatory bowel disease specialist nurses need to be educated about the potential benefits of acupuncture through publications and presentations at local, regional and national meetings. Partitioning with herbs or wheat ensured that moxa was not directly applied to the skin and, therefore, was again a form of indirect moxibustion and safe. Such

an approach was critical in the context of a study of Canadian gastroenterologists, which showed that almost half (49%) of respondents were uncomfortable discussing complementary therapies with their patients, with lack of knowledge being cited as the most common reason. Indeed, most (79%) had received no formal education with regard to such treatments [37]. The responsibility for the development of appropriate educational programs must lie with professional organisations, such as the British Medical Acupuncture Society.

**Author Contributions:** Conceptualisation, methodology, formal analysis, writing and editing of all drafts together with review of the published document were equally undertaken by A.F. and J.F.M. All authors have read and agreed to the published version of the manuscript.

**Funding:** This research received no external funding.

**Institutional Review Board Statement:** No institutional review was required as this is a review of published work, which is openly available.

**Informed Consent Statement:** The photographs are of the authors, who were not patients and both consented to their use. Ethical approval was given by the Leicester Complementary Therapy Centre at which the procedure was performed.

**Data Availability Statement:** This review is based on published available studies.

**Acknowledgments:** We would like to thank Mike Cummings for the preparation and provision of the two schemas of acupuncture points on the abdomen and leg. He is the Medical Director of the British Medical Acupuncture Society (BMAS).

**Conflicts of Interest:** Affifa Farrukh is a Counsel Member and Trustee of the British Medical Acupuncture Society.

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
