# Peer review of "Scoping Review of Acupuncture and Moxibustion in the Treatment of Inflammatory Bowel Disease"

_gastrointestdisord, doi:10.3390/gidisord5010010_

Round 1

Reviewer 1 Report

  • A brief summary In general the idea is good, nice presentation but needs more intro with diagrams 
  • General concept comments 
    Article: I think the areas of weakness include the proof of evidence base facts and a more detailed intro about the procedure itself with pictures or illustrations.
  • Specific comments The number of new citations is only 7 out of 38, I would recommend getting more new citations 

Author Response

Thank you for the comments.

We have now expanded description of the procedures and included photographs.

The reason that the number of new references is limited is that an extensive review using both PubMed and Google did not identify more recent papers. Recent work has concentrated on animal models and as explained in the text these were excluded because of the lack of clinical relevance of such models

Reviewer 2 Report

This is a well-written manuscript in which the authors describe an overview of the field of acupuncture in the treatment of inflammatory bowel disease with a view to providing a treatment option for patients with inflammatory bowel disease. I would suggest that the authors add a table with an overview of all clinical trials of acupuncture for IBD to facilitate the authors' reading and understanding.

Author Response

Thank you for the review

We have added a table summarising the clinical studies

Round 2

Reviewer 1 Report

This version after the modifications is much better, will clear take-home message 

Author Response

This reviewer has required no further changes

Reviewer 2 Report

Table 2 needs to be added to the literatures; ST25 in Fig. 1 is not shown. The position of the acupuncture points in Figures 2 and 3 is incorrect. The authors could have provided schematics rather than pictures of the actual treatment.

Author Response

Reference to Table 2 is included twice in the text Schematic drawings have been added (the other reviewer requested photographs) The positions are described in the drawings with traditional Chinese measurements. There can be discussion between acupuncturists and the sites, depending upon the school of acupuncture. These points are correctly identified.